# A Joint Many-Task Model: Growing a Neural Network for Multiple NLP Tasks

**Kazuma Hashimoto,**[*] **Caiming Xiong,**[†] **Yoshimasa Tsuruoka & Richard Socher**
The University of Tokyo
{hassy, tsuruoka}@logos.t.u-tokyo.ac.jp
Salesforce Research
{cxiong, rsocher}@salesforce.com

## Abstract

Transfer and multi-task learning have traditionally focused on either a single source-target pair or very few, similar tasks. Ideally, the linguistic levels of morphology, syntax and semantics would benefit each other by being trained in a single model. We introduce such a joint many-task model together with a strategy for successively growing its depth to solve increasingly complex tasks. All layers include shortcut connections to both word representations and lower-level task predictions. We use a simple regularization term to allow for optimizing all model weights to improve one task's loss without exhibiting catastrophic interference of the other tasks. Our single end-to-end trainable model obtains state-of-the-art results on chunking, dependency parsing, semantic relatedness and textual entailment. It also performs competitively on POS tagging. Our dependency parsing layer relies only on a single feed-forward pass and does not require a beam search.

## 1 Introduction

The potential for leveraging multiple levels of representation has been demonstrated in a variety of ways in the field of Natural Language Processing (NLP). For example, Part-Of-Speech (POS) tags are used to train syntactic parsers. The parsers are used to improve higher-level tasks, such as natural language inference (Chen et al., 2016), relation classification (Socher et al., 2012), sentiment analysis (Socher et al., 2013; Tai et al., 2015), or machine translation (Eriguchi et al., 2016). However, higher level tasks are not usually able to improve lower level tasks, often because systems are pipelines and not trained end-to-end.

In deep learning, unsupervised word vectors are useful representations and often used to initialize recurrent neural networks for subsequent tasks (Pennington et al., 2014). However, not being jointly trained, deep NLP models have yet shown benefits from predicting many ($> 4$) increasingly complex linguistic tasks each at a successively deeper layer. Instead, existing models are often designed to predict different tasks either entirely separately or at the same depth (Collobert et al., 2011), ignoring linguistic hierarchies.

We introduce a Joint Many-Task (JMT) model, outlined in Fig. 1, which predicts increasingly complex NLP tasks at successively deeper layers. Unlike traditional NLP pipeline systems, our single JMT model can be trained end-to-end for POS tagging, chunking, dependency parsing, semantic relatedness, and textual entailment. We propose an adaptive training and regularization strategy to grow this model in its depth. With the help of this strategy we avoid catastrophic interference between tasks, and instead show that both lower and higher level tasks benefit from the joint training. Our model is influenced by the observation of Søgaard & Goldberg (2016) who showed that predicting two different tasks is more accurate when performed in different layers than in the same layer (Collobert et al., 2011).

---

[*] Work was done while the first author was an intern at Salesforce Research.
[†] Corresponding author.

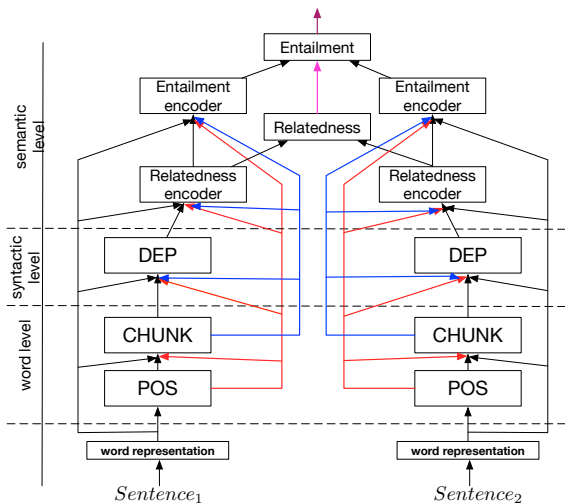

Figure 1: Overview of the joint many-task model predicting different linguistic outputs at successively deeper layers.

## 2 THE JOINT MANY-TASK MODEL

In this section, we assume that the model is trained and describe its inference procedure. We begin at the lowest level and work our way to higher layers and more complex tasks.

### 2.1 WORD REPRESENTATIONS

For each word $w_t$ in the input sentence $s$ of length $L$, we construct a representation by concatenating a word and a character embedding.

**Word embeddings:** We use Skip-gram (Mikolov et al., 2013) to train a word embedding matrix, which will be shared across all of the tasks. The words which are not included in the vocabulary are mapped to a special *UNK* token.

**Character $n$-gram embeddings:** Character $n$-gram embeddings are learned using the same skip-gram objective function as the word vectors. We construct the vocabulary of the character $n$-grams in the training data and assign an embedding for each character $n$-gram. The final character embedding is the average of the *unique* character $n$-gram embeddings of a word $w_t$.[1] For example, the character $n$-grams ($n = 1, 2, 3$) of the word "Cat" are {C, a, t, #BEGIN#C, Ca, at, t#END#, #BEGIN#Ca, Cat, at#END#}, where "#BEGIN#" and "#END#" represent the beginning and the end of each word, respectively. The use of the character $n$-gram embeddings efficiently provides morphological features and information about unknown words. The training procedure for the character $n$-gram embeddings is described in Section 3.1, and for further details, please see Appendix A. Each word is subsequently represented as $x_t$, the concatenation of its corresponding word and character vectors.

### 2.2 WORD-LEVEL TASK: POS TAGGING

The first layer of the model is a bi-directional LSTM (Graves & Schmidhuber, 2005; Hochreiter & Schmidhuber, 1997) whose hidden states are used to predict POS tags. We use the following Long Short-Term Memory (LSTM) units for the forward direction:

$$i_t = \sigma\left(W_i g_t + b_i\right), \qquad f_t = \sigma\left(W_f g_t + b_f\right), \qquad o_t = \sigma\left(W_o g_t + b_o\right),$$
$$u_t = \tanh\left(W_u g_t + b_u\right), \quad c_t = i_t \odot u_t + f_t \odot c_{t-1}, \quad h_t = o_t \odot \tanh\left(c_t\right), \tag{1}$$

---

[1]Wieting et al. (2016) used a nonlinearity, but we have observed that the simple averaging also works well.

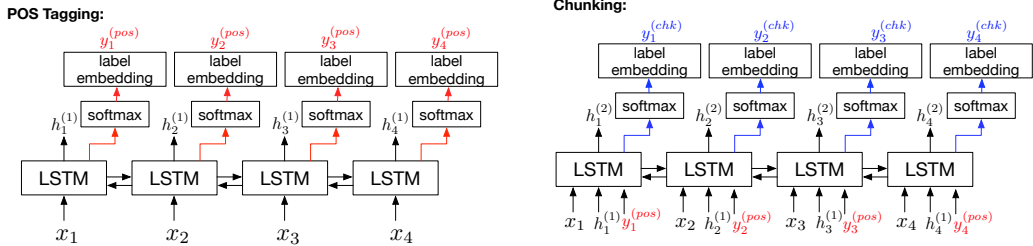

Figure 2: Overview of the POS tagging and chunking tasks in the first and second layers of the JMT model.

where we define the input $g_t$ as $g_t = [\overrightarrow{h}_{t-1}; x_t]$, i.e. the concatenation of the previous hidden state and the word representation of $w_t$. The backward pass is expanded in the same way, but a different set of weights are used.

For predicting the POS tag of $w_t$, we use the concatenation of the forward and backward states in a one-layer bi-LSTM layer corresponding to the $t$-th word: $h_t = [\overrightarrow{h}_t; \overleftarrow{h}_t]$. Then each $h_t$ ($1 \leq t \leq L$) is fed into a standard softmax classifier with a single ReLU layer which outputs the probability vector $y^{(1)}$ for each of the POS tags.

## 2.3 WORD-LEVEL TASK: CHUNKING

Chunking is also a word-level classification task which assigns a chunking tag (B-NP, I-VP, etc.) for each word. The tag specifies the region of major phrases (or chunks) in the sentence.

Chunking is performed in the second bi-LSTM layer on top of the POS layer. When stacking the bi-LSTM layers, we use Eq. (1) with input $g_t^{(2)} = [h_{t-1}^{(2)}; h_t^{(1)}; x_t; y_t^{(pos)}]$, where $h_t^{(1)}$ is the hidden state of the first (POS) layer. We define the weighted label embedding $y_t^{(pos)}$ as follows:

$$y_t^{(pos)} = \sum_{j=1}^{C} p(y_t^{(1)} = j | h_t^{(1)})\ell(j), \tag{2}$$

where $C$ is the number of the POS tags, $p(y_t^{(1)} = j | h_t^{(1)})$ is the probability value that the $j$-th POS tag is assigned to $w_t$, and $\ell(j)$ is the corresponding label embedding. The probability values are automatically predicted by the POS layer working like a built-in POS tagger, and thus no gold POS tags are needed. This output embedding can be regarded as a similar feature to the $K$-best POS tag feature which has been shown to be effective in syntactic tasks (Andor et al., 2016; Alberti et al., 2015). For predicting the chunking tags, we employ the same strategy as POS tagging by using the concatenated bi-directional hidden states $h_t^{(2)} = [\overrightarrow{h}_t^{(2)}; \overleftarrow{h}_t^{(2)}]$ in the chunking layer. We also use a single ReLU hidden layer before the classifier.

## 2.4 SYNTACTIC TASK: DEPENDENCY PARSING

Dependency parsing identifies syntactic relationships (such as an adjective modifying a noun) between pairs of words in a sentence. We use the third bi-LSTM layer on top of the POS and chunking layers to classify relationships between all pairs of words. The input vector for the LSTM includes hidden states, word representations, and the label embeddings for the two previous tasks: $g_t^{(3)} = [h_{t-1}^{(3)}; h_t^{(2)}; x_t; (y_t^{(pos)} + y_t^{(chk)})]$, where we computed the chunking vector in a similar fashion as the POS vector in Eq. (2). The POS and chunking tags are commonly used to improve dependency parsing (Attardi & DellOrletta, 2008).

Like a sequential labeling task, we simply predict the parent node (*head*) for each word in the sentence. Then a dependency label is predicted for each of the child-parent node pairs. To predict the parent node of the $t$-th word $w_t$, we define a matching function between $w_t$ and the candidates of the parent node as $m(t, j) = h_t^{(3)^{\mathrm{T}}} W_d h_j^{(3)}$, where $W_d$ is a parameter matrix. For the root, we

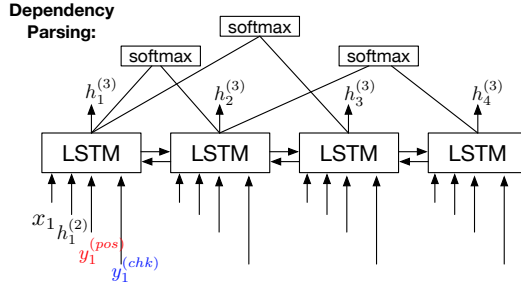

Figure 3: Overview of dependency parsing in the third layer of the JMT model.

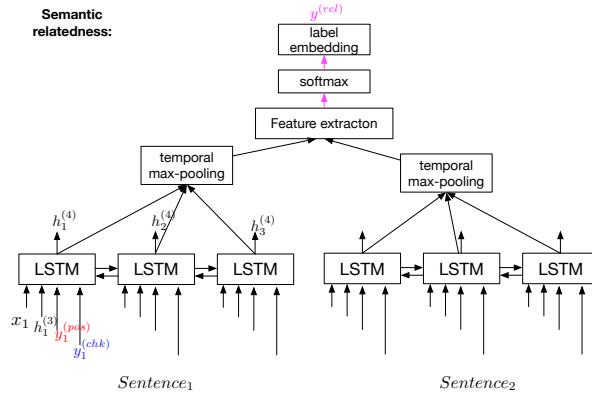

Figure 4: Overview of the semantic tasks in the top layers of the JMT model.

define $h_{L+1}^{(3)} = r$ as a parameterized vector. To compute the probability that $w_j$ (or the root node) is the parent of $w_t$, the scores are normalized:

$$p(j|h_t^{(3)}) = \frac{\exp\left(m\left(t, j\right)\right)}{\sum_{k=1, k \neq t}^{L+1} \exp\left(m\left(t, k\right)\right)},$$ 

(3)

where $L$ is the sentence length.

Next, the dependency labels are predicted using $[h_t^{(3)}; h_j^{(3)}]$ as input to a standard $\mathrm{softmax}$ classifier with a single $\mathrm{ReLU}$ layer. At test time, we greedily select the parent node and the dependency label for each word in the sentence.[2] At training time, we use the gold child-parent pairs to train the label predictor.

## 2.5  SEMANTIC TASK: SEMANTIC RELATEDNESS

The next two tasks model the semantic relationships between two input sentences. The first task measures the semantic relatedness between two sentences. The output is a real-valued relatedness score for the input sentence pair. The second task is a textual entailment task, which requires one to determine whether a premise sentence entails a hypothesis sentence. There are typically three classes: `entailment`, `contradiction`, and `neutral`.

The two semantic tasks are closely related to each other. If the semantic relatedness between two sentences is very low, they are unlikely to entail each other. Based on this intuition and to make use of the information from lower layers, we use the fourth and fifth bi-LSTM layer for the relatedness and entailment task, respectively.

---

[2]This method currently assumes that each word has only one parent node, but it can be expanded to handle multiple parent nodes, which leads to cyclic graphs.

Now it is required to obtain the sentence-level representation rather than the word-level representation $h_t^{(4)}$ used in the first three tasks. We compute the sentence-level representation $h_{\mathbf{s}}^{(4)}$ as the element-wise maximum values across all of the word-level representations in the fourth layer:

$$h_{\mathbf{s}}^{(4)} = \max\left(h_1^{(4)}, h_2^{(4)}, \ldots, h_L^{(4)}\right). \tag{4}$$

To model the semantic relatedness between $s$ and $s'$, we follow Tai et al. (2015). The feature vector for representing the semantic relatedness is computed as follows:

$$d_1(s, s') = \left[\left|h_{\mathbf{s}}^{(4)} - h_{\mathbf{s'}}^{(4)}\right|; h_{\mathbf{s}}^{(4)} \odot h_{\mathbf{s'}}^{(4)}\right], \tag{5}$$

where $\left|h_{\mathbf{s}}^{(4)} - h_{\mathbf{s'}}^{(4)}\right|$ is the absolute values of the element-wise subtraction, and $h_{\mathbf{s}}^{(4)} \odot h_{\mathbf{s'}}^{(4)}$ is the element-wise multiplication. Both of them can be regarded as two different similarity metrics of the two vectors. Then $d_1(s, s')$ is fed into a softmax classifier with a single Maxout hidden layer (Goodfellow et al., 2013) to output a relatedness score (from 1 to 5 in our case) for the sentence pair.

### 2.6 Semantic Task: Textual entailment

For entailment classification between two sentences, we also use the max-pooling technique as in the semantic relatedness task. To classify the premise-hypothesis pair $(s, s')$ into one of the three classes, we compute the feature vector $d_2(s, s')$ as in Eq. (5) except that we do not use the absolute values of the element-wise subtraction, because we need to identify which is the premise (or hypothesis). Then $d_2(s, s')$ is fed into a standard softmax classifier.

To make use of the output from the relatedness layer directly, we use the label embeddings for the relatedness task. More concretely, we compute the class label embeddings for the semantic relatedness task similar to Eq. (2). The final feature vectors that are concatenated and fed into the entailment classifier are the weighted relatedness label embedding and the feature vector $d_2(s, s')$.[3] We use three Maxout hidden layers before the classifier.

## 3 Training the JMT Model

The model is trained jointly over all datasets. During each epoch, the optimization iterates over each full training dataset in the same order as the corresponding tasks described in the modeling section.

### 3.1 Pre-Training Word Representations

We pre-train word embeddings using the Skip-gram model with negative sampling (Mikolov et al., 2013). We also pre-train the character $n$-gram embeddings using Skip-gram. The only difference is that each input word embedding in the Skip-gram model is replaced with its corresponding average embedding of the character $n$-gram embeddings described in Section 2.1. These embeddings are fine-tuned during the training of our JMT model. We denote the embedding parameters as $\theta_e$.

### 3.2 Training the POS Layer

Let $\theta_{\mathrm{POS}} = (W_{\mathrm{POS}}, b_{\mathrm{POS}}, \theta_e)$ denote the set of model parameters associated with the POS layer, where $W_{\mathrm{POS}}$ is the set of the weight matrices in the first bi-LSTM and the classifier, and $b_{\mathrm{POS}}$ is the set of the bias vectors. The objective function to optimize $\theta_{\mathrm{POS}}$ is defined as follows:

$$J_1(\theta_{\mathrm{POS}}) = -\sum_s \sum_t \log p\left(y_t^{(1)} = \alpha | h_t^{(1)}\right) + \lambda \|W_{\mathrm{POS}}\|^2 + \delta \|\theta_e - \theta_e'\|^2, \tag{6}$$

where $p(y_t^{(1)} = \alpha_{w_t} | h_t^{(1)})$ is the probability value that the correct label $\alpha$ is assigned to $w_t$ in the sentence $s$, $\lambda \|W_{\mathrm{POS}}\|^2$ is the L2-norm regularization term, and $\lambda$ is a hyperparameter.

---

[3]This modification does not affect the LSTM transitions, and thus it is still possible to add other single-sentence-level tasks on top of our model.

We call the second regularization term $\delta\|\theta_e - \theta'_e\|^2$ a *successive* regularization term. The successive regularization is based on the idea that we do not want the model to forget the information learned for the other tasks. In the case of POS tagging, the regularization is applied to $\theta_e$, and $\theta'_e$ is the embedding parameter after training the final task in the top-most layer at the previous training epoch. $\delta$ is a hyperparameter.

### 3.3 TRAINING THE CHUNKING LAYER

The objective function is defined as follows:

$$J_2(\theta_{\text{chk}}) = -\sum_s \sum_t \log p(y_t^{(2)} = \alpha|h_t^{(2)})d + \lambda\|W_{\text{chk}}\|^2 + \delta\|\theta_{\text{POS}} - \theta'_{\text{POS}}\|^2, \qquad (7)$$

which is similar to that of POS tagging, and $\theta_{\text{chk}}$ is $(W_{\text{chk}}, b_{\text{chk}}, E_{\text{POS}}, \theta_e)$, where $W_{\text{chk}}$ and $b_{\text{chk}}$ are the weight and bias parameters including those in $\theta_{\text{POS}}$, and $E_{\text{POS}}$ is the set of the POS label embeddings. $\theta'_{\text{POS}}$ is the one after training the POS layer at the current training epoch.

### 3.4 TRAINING THE DEPENDENCY LAYER

The objective function is defined as follows:

$$J_3(\theta_{\text{dep}}) = -\sum_s \sum_t \log p(\alpha|h_t^{(3)})p(\beta|h_t^{(3)}, h_\alpha^{(3)}) + \lambda(\|W_{\text{dep}}\|^2 + \|W_d\|^2) + \delta\|\theta_{\text{chk}} - \theta'_{\text{chk}}\|^2, \quad (8)$$

where $p(\alpha|h_t^{(3)})$ is the probability value assigned to the correct parent node $\alpha$ for $w_t$, and $p(\beta|h_t^{(3)}, h_\alpha^{(3)})$ is the probability value assigned to the correct dependency label $\beta$ for the child-parent pair $(w_t, \alpha)$. $\theta_{\text{dep}}$ is defined as $(W_{\text{dep}}, b_{\text{dep}}, W_d, r, E_{\text{POS}}, E_{\text{chk}}, \theta_e)$, where $W_{\text{dep}}$ and $b_{\text{dep}}$ are the weight and bias parameters including those in $\theta_{\text{chk}}$, and $E_{\text{chk}}$ is the set of the chunking label embeddings.

### 3.5 TRAINING THE RELATEDNESS LAYER

Following Tai et al. (2015), the objective function is defined as follows:

$$J_4(\theta_{\text{rel}}) = \sum_{(s,s')} \text{KL}\left(\hat{p}(s,s')\Big\|p(h_s^{(4)}, h_{s'}^{(4)})\right) + \lambda\|W_{\text{rel}}\|^2 + \delta\|\theta_{\text{dep}} - \theta'_{\text{dep}}\|^2, \qquad (9)$$

where $\hat{p}(s, s')$ is the gold distribution over the defined relatedness scores, $p(h_s^{(4)}, h_{s'}^{(4)})$ is the predicted distribution given the the sentence representations, and $\text{KL}\left(\hat{p}(s,s')\Big\|p(h_s^{(4)}, h_{s'}^{(4)})\right)$ is the KL-divergence between the two distributions. $\theta_{\text{rel}}$ is defined as $(W_{\text{rel}}, b_{\text{rel}}, E_{\text{POS}}, E_{\text{chk}}, \theta_e)$.

### 3.6 TRAINING THE ENTAILMENT LAYER

The objective function is defined as follows:

$$J_5(\theta_{\text{ent}}) = -\sum_{(s,s')} \log p(y_{(s,s')}^{(5)} = \alpha|h_s^{(5)}, h_{s'}^{(5)}) + \lambda\|W_{\text{ent}}\|^2 + \delta\|\theta_{\text{rel}} - \theta'_{\text{rel}}\|^2, \qquad (10)$$

where $p(y_{(s,s')}^{(5)} = \alpha|h_s^{(5)}, h_{s'}^{(5)})$ is the probability value that the correct label $\alpha$ is assigned to the premise-hypothesis pair $(s, s')$. $\theta_{\text{ent}}$ is defined as $(W_{\text{ent}}, b_{\text{ent}}, E_{\text{POS}}, E_{\text{chk}}, E_{\text{rel}}, \theta_e)$, where $E_{\text{rel}}$ is the set of the relatedness label embeddings.

## 4 RELATED WORK

Many deep learning approaches have proven to be effective in a variety of NLP tasks and are becoming more and more complex. They are typically designed to handle single tasks, or some of them are designed as general-purpose models (Kumar et al., 2016; Sutskever et al., 2014) but applied to different tasks independently.

For handling multiple NLP tasks, multi-task learning models with deep neural networks have been proposed (Collobert et al., 2011; Luong et al., 2016), and more recently Søgaard & Goldberg (2016) have suggested that using different layers for different tasks is more effective than using the same layer in jointly learning closely-related tasks, such as POS tagging and chunking. However, the number of tasks was limited or they have very similar task settings like word-level tagging, and it was not clear how lower-level tasks could be also improved by combining higher-level tasks.

In the field of computer vision, some transfer and multi-task learning approaches have also been proposed (Li & Hoiem, 2016; Misra et al., 2016). For example, Misra et al. (2016) proposed a multi-task learning model to handle different tasks. However, they assume that each data sample has annotations for the different tasks, and do not explicitly consider task hierarchies.

Recently, Rusu et al. (2016) have proposed a progressive neural network model to handle multiple reinforcement learning tasks, such as Atari games. Like our JMT model, their model is also successively trained according to different tasks using different layers called columns in their paper. In their model, once the first task is completed, the model parameters for the first task are fixed, and then the second task is handled by adding new model parameters. Therefore, accuracy of the previously trained tasks is never improved. In NLP tasks, multi-task learning has the potential to improve not only higher-level tasks, but also lower-level tasks. Rather than fixing the pre-trained model parameters, our successive regularization allows our model to continuously train the lower-level tasks without significant accuracy drops.

## 5 EXPERIMENTAL SETTINGS

### 5.1 DATASETS

**POS tagging:** To train the POS tagging layer, we used the Wall Street Journal (WSJ) portion of Penn Treebank, and followed the standard split for the training (Section 0-18), development (Section 19-21), and test (Section 22-24) sets. The evaluation metric is the word-level accuracy.

**Chunking:** For chunking, we also used the WSJ corpus, and followed the standard split for the training (Section 15-18) and test (Section 20) sets as in the CoNLL 2000 shared task. We used Section 19 as the development set, following Søgaard & Goldberg (2016), and employed the IOBES tagging scheme. The evaluation metric is the F1 score defined in the shared task.

**Dependency parsing:** We also used the WSJ corpus for dependency parsing, and followed the standard split for the training (Section 2-21), development (Section 22), and test (Section 23) sets. We converted the treebank data to Stanford style dependencies using the version 3.3.0 of the Stanford converter. The evaluation metrics are the Unlabeled Attachment Score (UAS) and the Labeled Attachment Score (LAS), and punctuations are excluded for the evaluation.

**Semantic relatedness:** For the semantic relatedness task, we used the SICK dataset (Marelli et al., 2014), and followed the standard split for the training (SICK_train.txt), development (SICK_trial.txt), and test (SICK_test_annotated.txt) sets. The evaluation metric is the Mean Squared Error (MSE) between the gold and predicted scores.

**Textual entailment:** For textual entailment, we also used the SICK dataset and exactly the same data split as the semantic relatedness dataset. The evaluation metric is the accuracy.

### 5.2 TRAINING DETAILS

**Pre-training embeddings:** We used the `word2vec` toolkit to pre-train the word embeddings. We created our training corpus by selecting lowercased English Wikipedia text and obtained 100-dimensional Skip-gram word embeddings trained with the context window size 1, the negative sampling method (15 negative samples), and the sub-sampling method ($10^{-5}$ of the sub-sampling coefficient).[4] We also pre-trained the character $n$-gram embeddings using the same parameter settings with the case-sensitive Wikipedia text. We trained the character $n$-gram embeddings for $n = 1, 2, 3, 4$ in the pre-training step.

---

[4]It is empirically known that such a small window size in leads to better results on syntactic tasks than large window sizes. Moreover, we have found that such word embeddings work well even on the semantic tasks.

**Embedding initialization:** We used the pre-trained word embeddings to initialize the word embeddings, and the word vocabulary was built based on the training data of the five tasks. All words in the training data were included in the word vocabulary, and we employed the *word-dropout* method (Kiperwasser & Goldberg, 2016) to train the word embedding for the unknown words. We also built the character $n$-gram vocabulary for $n = 2, 3, 4$, following Wieting et al. (2016), and the character $n$-gram embeddings were initialized with the pre-trained embeddings. All of the label embeddings were initialized with uniform random values in $[-\sqrt{6/(dim + C)}, \sqrt{6/(dim + C)}]$, where $dim = 100$ is the dimensionality of the label embeddings and $C$ is the number of labels.

**Weight initialization:** The dimensionality of the hidden layers in the bi-LSTMs was set to 100. We initialized all of the softmax parameters and bias vectors, except for the forget biases in the LSTMs, with zeros, and the weight matrix $W_d$ and the root node vector $r$ for dependency parsing were also initialized with zeros. All of the forget biases were initialized with ones. The other weight matrices were initialized with uniform random values in $[-\sqrt{6/(row + col)}, \sqrt{6/(row + col)}]$, where $row$ and $col$ are the number of rows and columns of the matrices, respectively.

**Optimization:** At each epoch, we trained our model in the order of POS tagging, chunking, dependency parsing, semantic relatedness, and textual entailment. We used mini-batch stochastic gradient decent to train our model. The mini-batch size was set to 25 for POS tagging, chunking, and the SICK tasks, and 15 for dependency parsing. We used a gradient clipping strategy with growing clipping values for the different tasks; concretely, we employed the simple function: $\min(3.0, depth)$, where $depth$ is the number of bi-LSTM layers involved in each task, and 3.0 is the maximum value. The learning rate at the $k$-th epoch was set to $\frac{\varepsilon}{1.0+\rho(k-1)}$, where $\varepsilon$ is the initial learning rate, and $\rho$ is the hyperparameter to decrease the learning rate. We set $\varepsilon$ to 1.0 and $\rho$ to 0.3. At each epoch, the same learning rate was shared across all of the tasks.

**Regularization:** We set the regularization coefficient to $10^{-6}$ for the LSTM weight matrices, $10^{-5}$ for the weight matrices in the classifiers, and $10^{-3}$ for the successive regularization term excluding the classifier parameters of the lower-level tasks, respectively. The successive regularization coefficient for the classifier parameters was set to $10^{-2}$. We also used *dropout* (Hinton et al., 2012). The dropout rate was set to 0.2 for the vertical connections in the multi-layer bi-LSTMs (Pham et al., 2014), the word representations and the label embeddings of the entailment layer, and the classifier of the POS tagging, chunking, dependency parsing, and entailment. A different dropout rate of 0.4 was used for the word representations and the label embeddings of the POS, chunking, and dependency layers, and the classifier of the relatedness layer.

# 6 RESULTS AND DISCUSSION

## 6.1 SUMMARY OF MULTI-TASK RESULTS

Table 1 shows our results of the test sets on the five different tasks.[5] The column "Single" shows the results of handling each task separately using single-layer bi-LSTMs, and the column "JMT$_{\text{all}}$" shows the results of our JMT model. The single task settings only use the annotations of their own tasks. For example, when treating dependency parsing as a single task, the POS and chunking tags are *not* used. We can see that all results of the five different tasks are improved in our JMT model, which shows that our JMT model can handle the five different tasks in a single model. Our JMT model allows us to access arbitrary information learned from the different tasks. If we want to use the model just as a POS tagger, we can use the output from the first bi-LSTM layer. The output can be the weighted POS label embeddings as well as the discrete POS tags.

Table 1 also shows the results of three subsets of the different tasks. For example, in the case of "JMT$_{\text{ABC}}$", only the first three layers of the bi-LSTMs are used to handle the three tasks. In the case of "JMT$_{\text{DE}}$", only the top two layers are used just as a two-layer bi-LSTM by omitting all information from the first three layers. The results of the closely-related tasks show that our JMT model improves not only the high-level tasks, but also the low-level tasks.

---

[5]The development and test sentences of the chunking dataset are included in the dependency parsing dataset, although our model does not explicitly use the chunking annotations of the development and test data. In such cases, we show the results in parentheses.

| | | Single | JMT$_{all}$ | JMT$_{AB}$ | JMT$_{ABC}$ | JMT$_{DE}$ |
|---|---|---|---|---|---|---|
| A | POS | 97.45 | 97.55 | 97.52 | 97.54 | n/a |
| B | Chunking | 95.02 | (97.12) | 95.77 | (97.28) | n/a |
| C | Dependency UAS | 93.35 | 94.67 | n/a | 94.71 | n/a |
| | Dependency LAS | 91.42 | 92.90 | n/a | 92.92 | n/a |
| D | Relatedness | 0.247 | 0.233 | n/a | n/a | 0.238 |
| E | Entailment | 81.8 | 86.2 | n/a | n/a | 86.8 |

Table 1: Test set results for the five tasks. In the relatedness task, the lower scores are better.

| Method | Acc. |
|---|---|
| JMT$_{all}$ | 97.55 |
| Ling et al. (2015) | **97.78** |
| Kumar et al. (2016) | 97.56 |
| Ma & Hovy (2016) | 97.55 |
| Søgaard (2011) | 97.50 |
| Collobert et al. (2011) | 97.29 |
| Tsuruoka et al. (2011) | 97.28 |
| Toutanova et al. (2003) | 97.27 |

Table 2: POS tagging results.

| Method | F1 |
|---|---|
| JMT$_{AB}$ | **95.77** |
| Søgaard & Goldberg (2016) | 95.56 |
| Suzuki & Isozaki (2008) | 95.15 |
| Collobert et al. (2011) | 94.32 |
| Kudo & Matsumoto (2001) | 93.91 |
| Tsuruoka et al. (2011) | 93.81 |

Table 3: Chunking results.

| Method | UAS | LAS |
|---|---|---|
| JMT$_{all}$ | **94.67** | **92.90** |
| Single | 93.35 | 91.42 |
| Andor et al. (2016) | 94.61 | 92.79 |
| Alberti et al. (2015) | 94.23 | 92.36 |
| Weiss et al. (2015) | 93.99 | 92.05 |
| Dyer et al. (2015) | 93.10 | 90.90 |
| Bohnet (2010) | 92.88 | 90.71 |

Table 4: Dependency results.

| Method | MSE |
|---|---|
| JMT$_{all}$ | **0.233** |
| JMT$_{DE}$ | 0.238 |
| Zhou et al. (2016) | 0.243 |
| Tai et al. (2015) | 0.253 |

Table 5: Semantic relatedness results.

| Method | Acc. |
|---|---|
| JMT$_{all}$ | 86.2 |
| JMT$_{DE}$ | **86.8** |
| Yin et al. (2016) | 86.2 |
| Lai & Hockenmaier (2014) | 84.6 |

Table 6: Textual entailment results.

## 6.2 COMPARISON WITH PUBLISHED RESULTS

**POS tagging:** Table 2 shows the results of POS tagging, and our JMT model achieves the score close to the state-of-the-art results. The best result to date has been achieved by Ling et al. (2015), which uses character-based LSTMs. Incorporating the character-based encoders into our JMT model would be an interesting direction, but we have shown that the simple pre-trained character $n$-gram embeddings lead to the promising result.

**Chunking:** Table 3 shows the results of chunking, and our JMT model achieves the state-of-the-art result. Søgaard & Goldberg (2016) proposed to jointly learn POS tagging and chunking in different layers, but they only showed improvement for chunking. By contrast, our results show that the low-level tasks are also improved by the joint learning.

**Dependency parsing:** Table 4 shows the results of dependency parsing by using only the WSJ corpus in terms of the dependency annotations, and our JMT model achieves the state-of-the-art result.[6] It is notable that our simple greedy dependency parser outperforms the previous state-of-the-art result which is based on beam search with global information. The result suggests that the bi-LSTMs efficiently capture global information necessary for dependency parsing. Moreover, our single task result already achieves high accuracy without the POS and chunking information. Further analysis on our dependency parser can be found in Appendix B.

**Semantic relatedness:** Table 5 shows the results of the semantic relatedness task, and our JMT model achieves the state-of-the-art result. The result of "JMT$_{DE}$" is already better than the previous state-of-the-art results. Both of Zhou et al. (2016) and Tai et al. (2015) explicitly used syntactic tree structures, and Zhou et al. (2016) relied on attention mechanisms. However, our method uses the simple max-pooling strategy, which suggests that it is worth investigating such simple methods before developing complex methods for simple tasks. Currently, our JMT model does not explicitly use the learned dependency structures, and thus the explicit use of the output from the dependency layer should be an interesting direction of future work.

---

[6]Choe & Charniak (2016) employed the tri-training technique to expand the training data with automatically-generated 400,000 trees in addition to the WSJ data, and they reported 95.9 UAS and 94.1 LAS.

**Textual entailment:** Table 6 shows the results of textual entailment, and our JMT model achieves the state-of-the-art result.[7] The previous state-of-the-art result in Yin et al. (2016) relied on attention mechanisms and dataset-specific data pre-processing and features. Again, our simple max-pooling strategy achieves the state-of-the-art result boosted by the joint training. These results show the importance of jointly handling related tasks. Error analysis can be found in Appendix C.

## 6.3 Analysis on Multi-Task Learning Architectures

Here, we first investigate the effects of using deeper layers for the five different single tasks. We then show the effectiveness of our training strategy: the successive regularization, the shortcut connections of the word representations, the embeddings of the output labels, the character $n$-gram embeddings, the use of the different layers for the different tasks, and the vertical connections of multi-layer bi-LSTMs. All of the results shown in this section are the development set results.

- **Depth:** The single task settings shown in Table 1 are obtained by using single layer bi-LSTMs, but in our JMT model, the higher-level tasks use successively deeper layers. To investigate the gap between the different number of the layers for each task, we also show the results of using multi-layer bi-LSTMs for the single task settings, in the column of "Single+" in Table 7. More concretely, we use the same number of the layers with our JMT model; for example, three layers are used for dependency parsing, and five layers are used for textual entailment. As shown in these results, deeper layers do not always lead to better results, and the joint learning is more important than making the models complex only for single tasks.

|  | Single | Single+ |
|---|---|---|
| POS | 97.52 | |
| Chunking | 95.65 | 96.08 |
| Dependency UAS | 93.38 | 93.88 |
| Dependency LAS | 91.37 | 91.83 |
| Relatedness | 0.239 | 0.665 |
| Entailment | 83.8 | 66.4 |

Table 7: Effects of depth for the *single* task settings.

- **Successive regularization:** In Table 8, the column of "w/o SR" shows the results of omitting the successive regularization terms described in Section 3. We can see that the accuracy of chunking is improved by the successive regularization, while other results are not affected so much. The chunking dataset used here is relatively small compared with other low-level tasks, POS tagging and dependency parsing. Thus, these results suggest that the successive regularization is effective when dataset sizes are imbalanced.

|  | $JMT_{all}$ | w/o SR |
|---|---|---|
| POS | 97.88 | 97.85 |
| Chunking | 97.59 | 97.13 |
| Dependency UAS | 94.51 | 94.46 |
| Dependency LAS | 92.60 | 92.57 |
| Relatedness | 0.236 | 0.239 |
| Entailment | 84.6 | 84.2 |

Table 8: Effectiveness of the Successive Regularization (SR).

- **Shortcut connections:** Our JMT model feeds the word representations into all of the bi-LSTM layers, which is called the shortcut connection. Table 9 shows the results of "$JMT_{all}$" with and without the shortcut connections. The results without the shortcut connections are shown in the column of "w/o SC". These results clearly show that the importance of the shortcut connections in our JMT model, and in particular, the semantic tasks in the higher layers strongly rely on the shortcut connections. That is, simply stacking the LSTM layers is not sufficient to handle a variety of NLP tasks in a single model. In Appendix D, we show how the shared word representations change according to each task (or layer).

|  | $JMT_{all}$ | w/o SC |
|---|---|---|
| POS | 97.88 | 97.79 |
| Chunking | 97.59 | 97.08 |
| Dependency UAS | 94.51 | 94.52 |
| Dependency LAS | 92.60 | 92.62 |
| Relatedness | 0.236 | 0.698 |
| Entailment | 84.6 | 75.0 |

Table 9: Effectiveness of the Shortcut Connections (SC).

- **Output label embeddings:** Table 10 shows the results without using the output labels of the POS, chunking, and relatedness layers, in the column of "w/o LE". These results show that the explicit use of the output information from the classifiers of the lower layers is important in our JMT model. The results in the column of "w/o SC&LE" are the ones without both of the shortcut connections and the label embeddings.

|  | $JMT_{all}$ | w/o LE | w/o SC&LE |
|---|---|---|---|
| POS | 97.88 | 97.85 | 97.87 |
| Chunking | 97.59 | 97.40 | 97.33 |
| Dependency UAS | 94.51 | 94.09 | 94.04 |
| Dependency LAS | 92.60 | 92.14 | 92.03 |
| Relatedness | 0.236 | 0.261 | 0.765 |
| Entailment | 84.6 | 81.6 | 71.2 |

Table 10: Effectiveness of the Label Embeddings (LE).

---

[7] The result of "$JMT_{all}$" is slightly worse than that of "$JMT_{DE}$", but the difference is not significant because the training data is small.

- **Character $n$-gram embeddings:** Table 11 shows the results for the three single tasks, POS tagging, chunking, and dependency parsing, with and without the pre-trained character $n$-gram embeddings. The column of "W&C" corresponds to using both of the word and character $n$-gram embeddings, and that of "Only W" corresponds to using only the word embeddings. These results clearly show that jointly using the pre-trained word and character $n$-gram embeddings is helpful in improving the results.

| Single | W&C | Only W |
|---|---|---|
| POS | 97.52 | 96.26 |
| Chunking | 95.65 | 94.92 |
| Dependency UAS | 93.38 | 92.90 |
| Dependency LAS | 91.37 | 90.44 |

Table 11: Effectiveness of the character $n$-gram embeddings.

The pre-training of the character $n$-gram embeddings is also effective; for example, without the pre-training, the POS accuracy drops from 97.52% to 97.38% and the chunking accuracy drops from 95.65% to 95.14%, but they are still better than those of using `word2vec` embeddings alone. Further analysis can be found in Appendix A.

- **Different layers for different tasks:** Table 12 shows the results for the three tasks of our "JMT$_{ABC}$" setting and that of not using the short-cut connections and the label embeddings as in Table 10. In addition, in the column of "All-3", we show the results of using the highest (i.e., the third) layer for all of the three tasks without any shortcut connections and label embeddings, and thus the two settings "w/o SC&LE" and "All-3" require exactly

| | JMT$_{ABC}$ | w/o SC&LE | All-3 |
|---|---|---|---|
| POS | 97.90 | 97.87 | 97.62 |
| Chunking | 97.80 | 97.41 | 96.52 |
| Dependency UAS | 94.52 | 94.13 | 93.59 |
| Dependency LAS | 92.61 | 92.16 | 91.47 |

Table 12: Effectiveness of using different layers for different tasks.

the same number of the model parameters. The results show that using the same layers for the three different tasks hampers the effectiveness of our JMT model, and the design of the model is much more important than the number of the model parameters.

- **Vertical connections:** Finally, we investigated our JMT results without using the vertical connections in the five-layer bi-LSTMs. More concretely, when constructing the input vectors $g_t$, we do not use the bi-LSTM hidden states of the previous layers. Table 13 shows the JMT$_{all}$ results with and without the vertical connections. As shown in the column of "w/o VC", we observed the competitive results. Therefore, in the target tasks used in our model, sharing the word representations and the output label embeddings is more effective than just stacking the bi-LSTM layers.

| | JMT$_{all}$ | w/o VC |
|---|---|---|
| POS | 97.88 | 97.82 |
| Chunking | 97.59 | 97.45 |
| Dependency UAS | 94.51 | 94.38 |
| Dependency LAS | 92.60 | 92.48 |
| Relatedness | 0.236 | 0.241 |
| Entailment | 84.6 | 84.8 |

Table 13: Effectiveness of the Vertical Connections (VC).

# 7 CONCLUSION

We presented a joint many-task model to handle a variety of NLP tasks with growing depth of layers in a single end-to-end deep model. Our model is successively trained by considering linguistic hierarchies, directly connecting word representations to all layers, explicitly using predictions in lower tasks, and applying successive regularization. In our experiments on five different types of NLP tasks, our single model achieves the state-of-the-art results on chunking, dependency parsing, semantic relatedness, and textual entailment.

ACKNOWLEDGMENTS

We thank the Salesforce Research team members for their fruitful comments and discussions.

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

## APPENDIX

## A    DETAILS OF CHARACTER $N$-GRAM EMBEDDINGS

Here we first describe the pre-training process of the character $n$-gram embeddings in detail and then show further analysis on the results in Table 11.

### A.1    PRE-TRAINING WITH SKIP-GRAM OBJECTIVE

We pre-train the character $n$-gram embeddings using the objective function of the Skip-gram model with negative sampling (Mikolov et al., 2013). We build the vocabulary of the character $n$-grams based on the training corpus, the case-sensitive English Wikipedia text. This is because such case-sensitive information is important in handling some types of words like named entities. Assuming that a word $w$ has its corresponding $K$ character $n$-grams $\{cn_1, cn_2, \ldots, cn_K\}$, where any overlaps and unknown ones are removed. Then the word $w$ is represented with an embedding $v_c(w)$ computed as follows:

$$v_c(w) = \frac{1}{K}\sum_{i=1}^{K} v(cn_i),\tag{11}$$

where $v(cn_i)$ is the parameterized embedding of the character $n$-gram $cn_i$, and the computation of $v_c(w)$ is exactly the same as the one used in our JMT model explained in Section 2.1.

The remaining part of the pre-training process is the same as the original Skip-gram model. For each word-context pair $(w, \overline{w})$ in the training corpus, $N$ negative context words are sampled, and the objective function is defined as follows:

$$\sum_{(w,\overline{w})} \left( -\log \sigma(v_c(w) \cdot \tilde{v}(\overline{w})) - \sum_{i=1}^{N} \log \sigma(-v_c(w) \cdot \tilde{v}(\overline{w}_i)) \right),\tag{12}$$

| Single (POS) | Overall Acc. | Acc. for unknown words |
|---|---|---|
| W&C | 97.52 | 90.68 (3,502/3,862) |
| Only W | 96.26 | 71.44 (2,759/3,862) |

Table 14: POS tagging scores on the development set with and without the character $n$-gram embeddings, focusing on accuracy for unknown words. The overall accuracy scores are taken from Table 11. There are 3,862 unknown words in the sentences of the development set.

| Single (Dependency) | Overall scores | | Scores for unknown words | |
|---|---|---|---|---|
| | UAS | LAS | UAS | LAS |
| W&C | 93.38 | 91.37 | 92.21 (900/976) | 87.81 (857/976) |
| Only W | 92.90 | 90.44 | 91.39 (892/976) | 81.05 (791/976) |

Table 15: Dependency parsing scores on the development set with and without the character $n$-gram embeddings, focusing on UAS and LAS for unknown words. The overall scores are taken from Table 11. There are 976 unknown words in the sentences of the development set.

where $\sigma(\cdot)$ is the logistic sigmoid function, $\tilde{v}(\overline{w})$ is the weight vector for the context word $\overline{w}$, and $\overline{w}_i$ is a negative sample. It should be noted that the weight vectors for the context words are parameterized for the words without any character information.

## A.2 Effectiveness on Unknown Words

One expectation from the use of the character $n$-gram embeddings is to better handle unknown words. We verified this assumption in the single task setting for POS tagging, based on the results reported in Table 11. Table 14 shows that the joint use of the word and character $n$-gram embeddings improves the score by about 19% in terms of the accuracy for unknown words.

We also show the results of the single task setting for dependency parsing in Table 15. Again, we can see that using the character-level information is effective, and in particular, the improvement of the LAS score is large. These results suggest that it is better to use not only the word embeddings, but also the character $n$-gram embeddings by default. Recently, the joint use of word and character information has proven to be effective in language modeling (Miyamoto & Cho, 2016), but just using the simple character $n$-gram embeddings is fast and also effective.

## B Analysis on Dependency Parsing

Our dependency parser is based on the idea of predicting a head (or parent) for each word, and thus the parsing results do not always lead to correct trees. To inspect this aspect, we checked the parsing results on the development set (1,700 sentences), using the "JMT$_{ABC}$" setting.

In the dependency annotations used in this work, each sentence has only one root node, and we have found 11 sentences with multiple root nodes and 11 sentences with no root nodes in our parsing results. We show two examples below:

  (a) Underneath the headline " Diversification , " it **counsels** , " Based on the events of the past week , all investors **need** to know their portfolios are balanced to help protect them against the market 's volatility . "

  (b) Mr. Eskandarian , who resigned his Della Femina post in September , becomes chairman and chief executive of Arnold .

In the example (a), the two boldfaced words "counsels" and "need" are predicted as child nodes of the root node, and the underlined word "counsels" is the correct one based on the gold annotations. This example sentence (a) consists of multiple internal sentences, and our parser misunderstood that both of the two verbs are the heads of the sentence.

In the example (b), none of the words is connected to the root node, and the correct child node of the root is the underlined word "chairman". Without the internal phrase "who resigned... in September",

the example sentence (b) is very simple, but we have found that such a simplified sentence is still not parsed correctly. In many cases, verbs are linked to the root nodes, but sometimes other types of words like nouns can be the candidates. In our model, the single parameterized vector $r$ is used to represent the root node for each sentence. Therefore, the results of the examples (a) and (b) suggest that it would be needed to capture various types of root nodes, and using sentence-dependent root representations would lead to better results in future work.

## C  ANALYSIS ON SEMANTIC TASKS

We inspected the development set results on the semantic tasks using the "JMT$_{\text{all}}$" setting. In our model, the highest-level task is the textual entailment task. We show an example premise-hypothesis pair which is misclassified in our results:

> Premise: "A surfer is riding a *big* wave across dark green water", and
>
> Hypothesis: "The surfer is riding a *small* wave".

The predicted label is `entailment`, but the gold label is `contradiction`. This example is very easy by focusing on the difference between the two words "big" and "small". However, our model fails to correctly classify this example because there are few opportunities to learn the difference. Our model relies on the pre-trained word embeddings based on word co-occurrence statistics (Mikolov et al., 2013), and it is widely known that such co-occurrence-based embeddings can rarely discriminate between antonyms and synonyms (Ono et al., 2015). Moreover, the other four tasks in our JMT model do not explicitly provide the opportunities to learn such semantic aspects. Even in the training data of the textual entailment task, we can find only one example to learn the difference between the two words, which is not enough to obtain generalization capacities. Therefore, it is worth investigating the explicit use of external dictionaries or the use of pre-trained word embeddings learned with such dictionaries (Ono et al., 2015), to see whether our JMT model is further improved not only for the semantic tasks, but also for the low-level tasks.

## D  HOW DO SHARED EMBEDDINGS CHANGE

In our JMT model, the word and character $n$-gram embedding matrices are shared across all of the five different tasks. To better qualitatively explain the importance of the shortcut connections shown in Table 9, we inspected how the shared embeddings change when fed into the different bi-LSTM layers. More concretely, we checked closest neighbors in terms of the cosine similarity for the word representations before and after fed into the forward LSTM layers. In particular, we used the corresponding part of $W_u$ in Eq. (1) to perform linear transformation of the input embeddings, because $u_t$ directly affects the hidden states of the LSTMs. Thus, this is a context-independent analysis.

Table 16 shows the examples of the word "standing". The row of "Embedding" shows the cases of using the shared embeddings, and the others show the results of using the linear-transformed embeddings. In the column of "Only word", the results of using only the word embeddings are shown. The closest neighbors in the case of "Embedding" capture the semantic similarity, but after fed into the POS layer, the semantic similarity is almost washed out. This is not surprising because it is sufficient to cluster the words of the same POS tags: here, `NN`, `VBG`, etc. In the chunking layer, the similarity in terms of verbs is captured, and this is because it is sufficient to identify the coarse chunking tags: here, `VP`. In the dependency layer, the closest neighbors are adverbs, gerunds of verbs, and nouns, and all of them can be child nodes of verbs in dependency trees. However, this information is not sufficient in further classifying the dependency labels. Then we can see that in the column of "Word and char", jointly using the character $n$-gram embeddings adds the morphological information, and as shown in Table 11, the LAS score is substantially improved.

In the case of semantic tasks, the projected embeddings capture not only syntactic, but also semantic similarities. These results show that different tasks need different aspects of the word similarities, and our JMT model efficiently transforms the shared embeddings for the different tasks by the simple linear transformation. Therefore, without the shortcut connections, the information about the word representations are fed into the semantic tasks after transformed in the lower layers where the

|  | Word and char | Only word |
|---|---|---|
| Embedding | leaning<br>kneeling<br>saluting<br>clinging<br>railing | stood<br>stands<br>sit<br>pillar<br>cross-legged |
| POS | warning<br>waxing<br>dunking<br>proving<br>tipping | ladder<br>rc6280<br>bethle<br>warning<br>f-a-18 |
| Chunking | applauding<br>disdaining<br>pickin<br>readjusting<br>reclaiming | fight<br>favor<br>pick<br>rejoin<br>answer |
| Dependency | guaranteeing<br>resting<br>grounding<br>hanging<br>hugging | patiently<br>hugging<br>anxiously<br>resting<br>disappointment |
| Relatedness | stood<br>stands<br>unchallenged<br>notwithstanding<br>judging | stood<br>unchallenged<br>stands<br>beside<br>exists |
| Entailment | nudging<br>skirting<br>straddling<br>contesting<br>footing | beside<br>stands<br>pillar<br>swung<br>ovation |

Table 16: Closest neighbors of the word "standing" in the embedding space and the projected space in each forward LSTM.

semantic similarities are not always important. Indeed, the results of the semantic tasks are very poor without the shortcut connections.

