# Peer review of "A Joint Many-Task Model: Growing a Neural Network for Multiple NLP Tasks"

_ICLR 2017 — rejected_

[Official Review · AnonReviewer3 · rating 3 · confidence 4 · 17 Dec 2016]
**experimental setup should be improved**

this work investigates a joint learning setup where tasks are stacked based on their complexity. to this end, experimental evaluation is done on pos tagging, chunking, dependency parsing, semantic relatedness, and textual entailment. the end-to-end model improves over models trained solely on target tasks.

although the hypothesis of this work is an important one, the experimental evaluation lacks thoroughness:

first, a very simple multi-task learning baseline [1] should be implemented where there is no hierarchy of tasks to test the hypothesis of the tasks should be ordered in terms of complexity.

second, since the test set of chunking is included in training data of dependency parsing, the results related to chunking with JMT_all are not informative. 

third, since the model does not guarantee well-formed dependency trees, thus, results in table 4 are not fair. 

minor issue:
- chunking is not a word-level task although the annotation is word-level. chunking is a structured prediction task where we would like to learn a structured annotation over a sequence [2].

[1]

[Official Review · AnonReviewer1 · rating 5 · confidence 4 · 17 Dec 2016]

The paper introduce a way to train joint models for many NLP tasks. Traditionally, we treat these tasks as “pipeline” — the later tasks will depending on the output of the previous tasks. Here, the authors propose a neural approach which includes all the tasks in one single model. The higher level tasks takes (1) the predictions from the lower level tasks and (2) the hidden representations of the lower level tasks. Also proposed in this paper, is the successive regularization. Intuitively, this means that, when training the high level tasks, we don’t want to change the model in the lower levels by too much so that the lower level tasks can keep a reasonable accuracy of prediction.

On the modeling side, I think the proposed model is very similar comparing to (Zhang and Weiss, ACL 2016) and SPINN (Bowman et al, 2016) in a even simpler way. The number of the experiments are good. But I am not sure I am convinced by the numbers in Table 1 since the patterns are not very clear there — sometimes, the performance of the higher level tasks even goes down when training with more tasks (sometimes it does go up, but also not very significant and stable). The dependency scores, although I don’t think this is a serious problem, comparing the UAS/LAS when the output is not guaranteed to be a well-formed tree isn’t strictly speaking fair.

I admit that the successive regularization make sense intuitively and is a very interesting direction to try. However, without a careful study of the training schema of such model, the current results on successive regularization do not convince me that it should be the right thing to do in such models (the current results are not strong enough to show that). The training methods need to be explored here including things as iteratively train on different tasks, and the relationship between the number of training iterations of a task and it’s training set size (and loss on this task etc).

[Official Review · AnonReviewer2 · rating 6 · confidence 4 · 18 Dec 2016]
**A joint model that actually works, limited novelty, a lot of experiments but possibly missing few important points.**

The authors propose a transfer learning approach applied to a number of NLP tasks; the set of tasks appear to have an order in terms of complexity (from easy syntactic tasks to somewhat harder semantic tasks).

Novelty: the way the authors propose to do transfer learning is by plugging models corresponding to each task, in a way that respects the known hierarchy (in terms of NLP "complexity") of those tasks. In that respect, the overall architecture looks more like a cascaded architecture than a transfer learning one. There are some existing literature in the area (first two Google results found:

[Final Decision · Program Chairs · 06 Feb 2017]
**ICLR committee final decision**

There is a bit of spread in the reviewer scores, but ultimately the paper does not meet the high bar for acceptance to ICLR. The lack of author responses to the reviews does not help either.